# The Role and Application of Exosomes and Their Cargos in Reproductive Diseases: A Systematic Review

**DOI:** 10.3390/vetsci9120706

**Published:** 2022-12-19

**Authors:** Zhi Chen, Xiangguo Wang

**Affiliations:** 1College of Biological Science and Agriculture, Qiannan Normal University for Nationalities, Duyun 558000, China; 2Animal Science and Technology College, Beijing University of Agriculture, Beijing 102206, China

**Keywords:** exosomes, polycystic ovary syndrome, preeclampsia, premature ovarian failure, polycystic, endometriosis, reproductive cancer

## Abstract

**Simple Summary:**

Exosomes are a subtype of extracellular vesicles with a size range between 30 and 150 nm, which can be released by the majority of cell types and circulate in body fluids. Exosomes are composed of lipids, proteins, RNA, DNA, and other substances related to cell origin. Moreover, exosomes can use their small size and relative durability to cross various cellular barriers and reach target cells to play a regulatory role. Therefore, exosomes have emerged as a potential test target and potential treatment options for reproductive diseases, including polycystic ovary syndrome, preeclampsia, premature ovarian failure, polycystic, endometriosis, and reproductive cancers. In this review, a systematic search was conducted in PubMed using the related keywords through September 2022 and then highlights our knowledge about the contribution of exosomes their cargos to reproductive diseases, such as proteins, lipids, DNA, and RNA. In the future, with the development of various highly-sensitive and high-throughput analytical methods, the research and application of exosomes in the field of reproductive disease detection will be greatly accelerated.

**Abstract:**

In recent years, the incidence of the reproductive diseases is increasing year-by-year, leading to abortion or fetal arrest, which seriously affects the reproductive health of human beings and the reproductive efficiency of animals. Exosomes are phospholipid bilayer vesicles that are widely distributed in living organisms and released by the cells of various organs and tissues. Exosomes contain proteins, RNA, lipids, and other components and are important carriers of information transfer between cells, which play a variety of physiological and pathological regulatory functions. More and more studies have found that exosomes and their connotations play an important role in the diagnosis, prognosis and treatment of diseases. A systematic review was conducted in this manuscript and then highlights our knowledge about the diagnostic and therapeutic applications of exosomes to reproductive diseases, such as polycystic ovary syndrome (PCOS), endometriosis, premature ovarian failure (POF), preeclampsia, polycystic, endometrial cancer, cervical cancer, ovarian cancer, and prostate gland cancer.

## 1. Search Strategy

A systematic search was conducted in PubMed using the related keywords through September 2022. Related data from original researches and reviews were integrated to comprehensively review the current findings and understanding of exosomes and their cargos in reproductive diseases. The references cited in this manuscript are the in-depth analysis of exosomes and their inclusions that play a major role in reproductive diseases. In certain parts where human research was limited, data from animal studies was used for supplementation. The references are all scientific papers published in English.

## 2. Exosomes

Exosomes were first isolated from reticulocyte culture medium as vesicles with lipid bilayer structure, 30–150 nm in diameter [1]. Current studies have indicated that exosomes are widely found in blood and various body fluids. As a new membrane transmission system of information between cells, exosomes regulate the physiological state of cells by carrying and transmitting various signal molecules and are involved in the occurrence and development of a variety of diseases [2]. Cell communication mediates cell behavior, biomarker screening, and drug carrier development, which are the main fields of exosome research at present.

### 2.1. Formation of Exosomes

Exosomes from different cells have similar biosynthesis pathways, all of which originate from endocytosis (Figure 1). An early secretory endosome is formed with intracellular plasmic membrane invagination, which then buds inward and envelops proteins, nucleic acids, and other substances to further mature, forming intracavitary vesicles (ILVs) contained in the endosome, called multivesicular bodies (MVBs). Eventually, MVBs fuses with lysosomes, autophagosomes, or plasma membrane to release the ILVs included in MVBs (exosomes) [3]. Therefore, most proteins contained in exosomes are related to the endocytosis network, which is the basis of homogenization and standardized identification of exosomes. Endosomal membrane budding forms ILVs mainly through endosomal sorting complex required for transport (ESCRT) [4]. The ESCRT mechanism is a 4-protein complex composed of ESCRT-0, -I, -II, and -III; vacuolar protein sorting 4 (VPS4); VPS20 correlation 1 (VTA1) and ALIX; and 30 other proteins [5]. The ubiquitin binding subunit of ESCRT-0 recognizes and sequesters ubiquitin proteins to specific domains in endosomal membranes to initiate the ESCRT mechanism and subsequently recruit ESCRT-I and -II bound cargo. After interacting with the ESCRT-I and -II complexes, ESCRT-III then binds to the total complex and is formed through the plasma membrane budding, which is a protein complex involved in promoting the budding process. Finally, the buds are divided to form ILVs [6].

### 2.2. Multilaterality of Exosomal Contents

Exosomes contain DNA, RNA, proteins, and lipids, which can be transmitted to other cells as signaling molecules to change the function of other cells. Among them, proteins are the main components of exosomes. According to Vesiclep database, exosome vesicles contain about 92 897 proteins [7]. These proteins can be roughly divided into two categories; one is the proteins commonly present in exosomes and involved in the formation of exosome membrane structure, which are mostly distributed on the surface or lumen of exosomes, including cytoskeletal components such as tubulin; actin and microfilament-binding proteins; membrane transport and fusion-related proteins such as the Rab protein; Flotillin; Alix and TSG101; some signal molecules such as protein kinases; and members of the four-transmembrane region protein superfamily such as CD63, CD9, CD82, and CD81 are closely related to exosome production [8]. The other class of proteins is related to cellular origin and is relatively specific. The antigen-presenting cells-derived exosomes are rich in MHC-Ⅰ and MHC-ⅠL CD80 and CD86; the platelet-derived exosomes contain hemophilia factor and integrin CD41a; and the exosomes secreted by T lymphocytes contain perforin and granzyme on the surface. Overexpressed protein markers of tumor antigens and immunosuppressive proteins, including FasL, TGF-β, and TRAIL, can also be detected in exosomes produced by tumor cells. In addition, small RNA in exosomes, such as miRNAs, are not only endogenous regulators of genetic information, but also factors regulating cell communication and early diagnostic markers of diseases. Therefore, exosomes have important potential application value in the diagnosis of reproductive diseases.

## 3. Exosomes in Reproductive Diseases

All germ cells produce exosomes and are found in body fluids such as blood, urine, breast milk, and saliva. Exosomes have stable biological activity, are easy to preserve and obtain, and can fuse through various cellular barriers. The secretion of exosomes and the changes of inclusions reflect different reproductive physiological and pathological states [9]. Therefore, exosomes and their miRNAs and protein components have become important markers in the diagnosis of reproductive system diseases [9,10].

### 3.1. Premature Ovarian Failure

According to a survey, 1% of women between 30 and 39 years old suffer from POF, and the diagnosis rate is increasing year-by-year. The occurrence of POF is influenced by many factors [11]. Due to the lack of understanding of its clinical symptoms and those symptoms’ similarity to other diseases, timely diagnosis and etiological research on POF have become major challenges in endocrinology.

Studies have found that exosomes derived from stem cells are widely concerned in the field of POF therapy. Exosomal miR-144-5p derived from bone marrow mesenchymal stem cells (BMSCs) mitigated cyclophosphamide (CTX)-induced follicular atresia in rats and reduced apoptosis of GCs through the phosphatase and tensin homolog (PTEN) pathway [12]. BMSC-derived exosomal miR-644-5p inhibits apoptosis of ovarian granulosa cells by targeting the regulation of p53 [13]. Human amniotic epithelial cell (hAEC) derived exosomes inhibit chemotherapy-induced granulosa cell apoptosis by transferring functional miR-1246 [14]. Amniotic fluid stem cell (AFSC) derived exosomal miR-146a and miR-10a prevented ovarian follicular atresia in CTX-treated mice by targeting Bim, Irak1, and Traf6 genes [15].

### 3.2. Polycystic Ovarian Syndrome

Polycystic ovary syndrome (PCOS) is one of the most common metabolic and endocrine disorders in premenopausal women, affecting about 6–8% of women worldwide. Women with the disease are at increased risk of glucose intolerance and type 2 diabetes, as well as hypertension, dyslipidemia, low fertility, and obstetric complications; endometriosis or cancer; possible ovarian malignancies; and mood and psychosexual disorders [16].

The bioinformatics analysis of plasma exosomal RNA from PCOS patients and healthy women revealed a large number of differential miRNA and LncRNA (as shown in Table 1), among which these differential miRNAs regulate PCOS by MAPK signaling, axon guidance, circadian rhythm, endocytosis, and cancer pathways [17]. LncRNA can regulate the pathogenesis of PCOS by targeting endocytosis, Hippo signaling pathway, MAPK signaling pathway, and human T-cell lymphotropic virus type 1 (HTLV-1) infection. Differential miRNA and lncRNA may be potential targets for the diagnosis and treatment of PCOS [18].

For example, exosomal miR-18b-5p derived from follicular fluid (FF) inhibits the development of polycystic ovary syndrome by targeting the PTEN-mediated phosphoinositide 3-kinase (PI3K)/Akt/mTOR signaling pathway [19]. Results of another study showed that miR-143-3p was highly expressed in the FF-exosomes derived from PCOS patients and promoted apoptosis of granulation cells by targeting bone morphogenetic protein receptor type 1A (BMPR1A) [20,21]. In addition, exosomal miR-155-5p and miR-143-3p derived from FF antagonistically regulate glycol-mediated follicular dysplasia of GC in PCOS [21,22]. PCOS follicular fluid derived exosome miR-424-5p inhibits GC proliferation and induces cell senescence in PCOS by blocking cell division cycle associated 4 (CDCA4)-mediated Rb/E2F1 signaling [23]. Mesenchymal stem cell (AMSC) derived exosome miR-323-3p attenuates PCOS by targeting programmed cell death protein 4 (PDCD4) in PCOS [24]. In addition to miRNA, the exosomes derived from PCOS patients contain higher levels of S100-A9 protein. S100-a9-rich exosomes significantly enhance inflammation and disrupt stereogenesis by activating the NF-κB signaling pathway and inducing proinflammatory factor expression levels in steroid-induced human granulomatous tumor cell line (KGN) (Table 1) [25].

### 3.3. Preeclampsia

Preeclampsia (PE) is a major complication of pregnancy, affecting about 4–5% of pregnancies worldwide and causing 10–15% of fetal deaths [26,27]. The role of exosome inclusions (such as miRNAs) in the diagnosis and therapy of PE has been the focus of research and some achievements have been achieved.

Bioinformatic analysis of plasma exosomal miRNA from PE patients and healthy women revealed a large number of differential miRNAs, which are viable potential biomarkers for predicting the risk of PE. Many studies have also explored the regulatory role of miRNAs in the pathogenesis of PE (as shown in Table 1). For example, some studies have found that human umbilical cord MSCs (hucMSCs) derived exosomal miR-133b promoted the migration, invasion and proliferation of trophoblast cells in PE by inhibiting serum and glucocorticoid-regulated kinase 1 (SGK1) [28]. MiR-584c-5p down-regulates PE inflammation by targeting protein tyrosine phosphatase receptor type O (PTPRO) [29]. MiR-203a-3p in serum-derived exosomes inhibits PE inflammation by regulating IL-24 [30]. Exosomal miR-18b-3p derived from hucMSCs inhibits preeclampsia by targeting leptin (LEP) [31]. Exosomal miR-125a-5p derived from cord blood (UCB) inhibits the migration and proliferation of trophoblast cells (TCs) by regulating the expression of VEGFA in PE [32]. Human placental microvascular endothelial cells (HPVECs) derived exosomal miR-486-5p can regulate the proliferation and invasion of TCs by targeting IGF1 and Rho GTPase activating protein 5 (ARHGAP5) [33,34]. In addition to numerous miRNAs, pteromalus puparum 13 (PP13) is a polypeptide in the blood-derived exosomes of patients with preeclampsia. It is decreased in the early stage of preeclampsia and increased in the later stage and through regulating the release of prostacycline to reshape the maternal spiral artery of the early placenta [35].

### 3.4. Endometriosis

Endometriosis (EMS) is a chronic inflammation with estrogen dependence, which affects about 6–10% of women around the world. It can not only cause chronic pelvic pain in women, but also affect ovarian function to a certain extent [36,37]. With regard to the role of exosomes in the development of endometriosis, people initially focused on the potential role of exosomes as the biomarkers of endometriosis (as shown in Table 1) [38]. However, subsequent studies have found that exosomes can also play certain therapeutic roles in the treatment of endometriosis and have a certain influence on the development of the disease [39,40]. Proteins such as PRDX1, histone H2A 2-C, ANXA2, α trypsin inhibitor heavy chain H4 (ITIH4), and microtubulin α chain were found in the exosomes of peritoneal fluid in different cycles of EMS [41], opening up a new approach for the diagnosis and therapy of endometriosis.

The formation of nerves and blood vessels is a key process in the development of endometriosis [42]. Related studies have reported that when the secretion of exosomes is blocked, the function of promoting nerve and angiogenesis is reduced [43]. The miRNAs contained in exosomes play an important role in promoting neuroangiogenesis. Several reports have found that exosomes containing miR-21, TSP1/THBS1, and TSP2/THBS2 can stimulate cell angiogenesis in other systems [44,45,46]. Other studies have found that antisense hypoxia-inducible factor (aHIF) is highly expressed in the exosomes derived from patients serum with endometriosis, and the aHIF shuttled by exosomes is transferred from endometriotic stromal cell (ECSC) to HUVECs, which in turn activate vascular endothelial growth factor (VEGF-A), VEGF-D, and basic fibroblast growth factor inducing angiogenic behavior in HUVECs, thereby promoting endometriosis angiogenesis [47,48]. Other studies have found that exosomal HOX transcript antisense RNA (HOTAIR) promotes endometriosis development and angiogenesis by means of targeting the miR-761/HDAC1 axis and stat3-mediated inflammation in vitro and in vivo [49].

In recent years, exosomes have been found to improve endometriosis through the regulation of the immune response. For example, MiR-22-3p and miR-320a target and regulate TNF signaling [50]. Exosomes can also substitute to activate macrophages, polarizing them into m2-like phenotypes, reducing their phagocytosis, and thus promoting the development of the endometriosis [51]. Exosomal miR-214-3p inhibits the process of endometriosis fibrosis by regulating connective tissue growth factor (CCN2) [52,53]. In addition to miRNAs, the secretion and content of lncRNA CHl1-AS1 increased in exosomes derived from peritoneal macrophages (pMφ) leads to miR-610 down-regulation and MDM2 up-regulation, thereby promoting the migration invasion and proliferation of ectopic endometrial stromal cells (eESCs) and apoptosis-inhibited [54]. Under the condition of endometriosis, the up-regulation of the exosomal miR-301a-3p significantly enhanced the ability of exosomes to induce macrophage M2 transformation, promoted the expression of arginase-1 (ARG-1) and PI3K, and inhibited the expression of PTEN [55]. Exosomal tRF-Leu-AAG-001 promotes both inflammation and angiogenesis in the development of endometriosis [56].

### 3.5. Reproductive Cancers

Breast cancer (BC) is one of the most common female malignancies, and metastasis is the main cause of death [57,58]. As a key component of exosomes, glycoproteins are derived from cancer cells (CCs) and involved in the metastasis of breast cancer cells transmitted through exosomes [57]. Exosomes derived from BC can enhance the migration of cancer endothelial cells. In addition to facilitating the migration and invasion of BC cells, tumor exosomes can induce drug resistance, and circulating exosomes may inhibit the sensitivity of BC cells to tamoxifen, which is applied for treating this disease. Currently, exosome miRNAs are the main diagnostic marker factors for BC, such as miR-372, miR-128-2, and miR-373 et al. (as shown in Table 1) [59,60,61,62,63].

Ovarian cancer (OC) is considered one of the deadliest and most malignant cancer diseases in women in the world [64,65]. At present, human epididymis protein 4 (HE4) and carbohydrate antigen-125 (CA125) are two common application markers in the diagnosis of ovarian cancer, but they lack specificity and sensitivity [66]. An analysis of exosomes obtained from 21 patients serum with ovarian cancer showed that CD24 protein expression was significantly correlated with the development of OC, suggesting that CD24 protein may also be used as a biomarker of OC [66]. OC is characterized by chemoresistance through exosomes and secreting and carrying calcium-dependent multifunctional actin binding protein–plasma coagulation sol protein (pGSN), which may be applied for a marker of chemoresistance in OC cells [67]. Beyond those, there are a lot of signaling molecules shown in Table 1, such as circRNAs [68,69,70,71,72,73] and miRNAs [74,75]. In the diagnosis of OC, exosomal miRNA and circRNA analysis in biopsy and serum samples may also be a potential diagnostic method for OC [76].

Cervical cancer (CC) is a common gynecological cancer and human papillomavirus (HPV) virus was the main inducing factor [77]. In CC, the increased secretion of exosome promotes angiogenesis and metastasis [77]. In addition, the content and composition of exosomes in CC cells are also affected by oncogenes of HPV-E6 and -E7, among which epithelial tumor signaling pathway Hedgehog (Hh)-GLI may be a potential marker of CC [77]. In addition, miR-221-3p has been shown to be an important inducer of local angiogenesis, and its expression is associated with increased vascular density in tumor microcirculation [78,79]. The injection of CC derived exosomal miR-221-3p into a tumor mouse model increased the process of angiogenesis and promoted the development and growth of CC cells [78,79]. In addition, there are also many functional exosome molecules shown in Table 1, such as miRNAs [80,81] and lncRNAs [82,83].

Endometrial cancer (EC) is usually diagnosed at a later stage and often associated with malignancy. The early detecting of EC in the progression of the disease is essential for treatment effect. For example, as a glycoprotein, galectin-3 binding protein (LGALS3BP) is often found in nearly all bodily fluids [84]. Related studies have reported that the content of LGALS3BP protein in plasma exosomes of patients with endometrial cancer is elevated [84]. This study indicates that exosomal LGALS3BP may be applied as a diagnosing marker for EC. Exosome-derived miRNAs also play an important role in the development of EC by regulating cell proliferation [85,86,87,88,89,90]. In addition to numerous miRNAs, Endothelial (CD144+), Monocytic (CD14+), and Annexin A2 (ANXA2) correlates with high grade, non-endometrioid subtype, advanced stage, and increased risks of recurrence [91,92].

Prostate cancer (PC) is a cancer that only affects men’s health, and prostate-specific antigen (PSA) is a recognized diagnostic marker [51]. PSA has some limitations to distinguish between stages of PC due to this inability. Exosomes derived from men’s urine or semen applied for diagnostic and therapeutic purposes is an alternative screening method [51]. MiRNAs such as miR-223-3p, miR-142-3p, and miR-142-5p have been shown to be associated with the incidence of PC, and the detection of these miRNAs will contribute to faster diagnosis and prognosis of PC (as shown in Table 1) [93,94,95,96,97,98]. In addition, an analysis of urine derived exosomes may be helpful in evaluating protein biomarkers of kidney injury-aquaporin 1 or AMP-3 dependent circulating transcription factors. Exosomal ephrinA2 and serum ephrinA2 can be used to distinguish PC patients from benign prostatic hyperplasia (BPH) patents [99]. In summary, miRNAs and genes involved in the progression of PC that contained in urine-derived exosomes have been screened as PC prostate-specific antigen diagnosing biomarkers [5].

**Table 1 vetsci-09-00706-t001:** Exosomal markers for the diagnosis of reproductive diseases and their related functions.

Diseases	Biomarkers	Variation Trend	Related Function	Reference
Polycystic ovarian syndrome	miR-25-3p	Increased	Amino acid metabolic pathway	[20,21]
	miR-141-3p, miR-200a-3p, miR-200c-3p, miR-483-3p, miR-3911, miR-199a-5p, miR-199a-3p, miR-199b-3p, miR-629-5p, miR-193b-3p, miR-6087, miR-10a-5p, miR-23b-3p, miR-98-5p, miR-382-5p, miR-483-5p, miR-4532, miR-4745-3p, miR-143-3p	Decreased		
	miR-126-3p, miR-146a-5p	Increased	Endocytosis tumorigenesis pathways, axon guidance, circadian rhythm and MAPK signaling pathway	[17]
	miR-106a-5p, miR-20b-5p	Decreased		
	S100-A9	Increased	Enhances inflammation and disrupts steroid production by activating NF-κB pathway	[25]
	miR-424-5p	Decreased	Blocks CDCA4-mediated Rb/E2F1 signaling and inhibits the proliferation of primary granulosa cells (GC) and induces cell senescence in PCOS	[23]
	lncRNA-H19, lncRNA-POP4, lncRNA-AKT3, lncRNA-HDAC6, lncRNA-NF1, lncRNAMUM1, lncRNA-LINC00173, lncRNA-DICER1, lncRNA-PTEN	Increased	Endocytosis, hippo signaling pathway, MAPK signaling pathway and HTLV-1 infection	[18]
	miR-27a-5p	Increased	Promotes cell migration and invasion	[100]
Preeclampsia	has-miR-525-5p	Increased	Inhibits vasoactive intestinal peptide	[29,32,101]
	has-miR-526b-5p	Increased	MMP-1 and HIF-1A	
	miR-192-5p, miR-205-5p, miR-208a-3p, miR-335-5p, miR-451a, miR-518a-3p, miR-542-3p, miR-23a-3p, miR-125b-2-3p, miR-144-3p,	Decreased	Inhibits TC migration and proliferation by targeting VEGFA	
	miR-141-3p, miR-199a-3p, miR-221-3p, miR-584-5p, miR-744-5p, miR-6724-5p let-7a-5p, miR-17-5p, miR-26a-5p, miR-30c-5p	Increased	Down-regulated inflammation by targeting PTPRO	
	has-miR-370-3p	Increased	Inhibits the proliferation, migration and invasion and promotes the apoptosis of trophoblast cells	[102]
	miR-15a-5p	Increased	Inhibits proliferation, invasion and apoptosis of trophoblast cells by targeting CDK1	[103]
	miR-19a-3p, miR-19b-3p, miR-376c-3p	Decreased	unknown	[104]
	miR-885-5p	Increased	Related to the liver enzyme aspartate aminotransferase	
	miR-141, miR-133	Increased	Regulation of trophoblast invasion and intercellular communication	[28,105]
	PP13	Early decline, late rise	Increased prostacyclin release for remodeling of maternal spiral arteries in early placenta	[35]
Endometriosis	miR-22-3p, miR-320a	Increased	TNF signaling, thyroid cancer, terpenoid skeleton biosynthesis, regulating stem cell pluripotency	[50]
	miR-214	Increased	Inhibit fibrosis and regulate the development of endometriosis lesions	[52,53]
	Thrombospondin-1 (TSP1/THBS1), TSP2/THBS2, Pigment epithelium-derived factor (SERPINF1), angiopoietin-related protein 6	Increased	Angiogenesis, immune system regulation and metabolic process pathways	[46]
	PRDX1	Increased	Protooncogene	[41]
	H2A type 2-C		Unknown	
	ANXA2		Promote tumor metastasis	
	ITIH4		It was expressed during the surgical trauma period	
	Tubulin α-chain		Unknown	
	miR-26b-5p, miR-215-5p	Decreased	Involves in MAPK and PI3K-Akt signaling pathways	[106]
	miR-6795-3p	Increased		
	lncRNA aHIF	Increased	Promotes angiogenesis in endometriosis	[47,48]
	lncRNA CHL1-AS1	Increased	Promotes the migration, invasion and proliferation of endometrial stromal cells (eESC) and inhibits apoptosis	[54]
	tRF-Leu-AAG-001	Increased	Promotes inflammation and angiogenesis	[56]
	miR-30c	Decreased	Reduces the invasion and migration of ectodermal endometrial epithelial cells (EECs) by targeting BCL9	[107]
	B-cell lymphoma 9 (BCL9)	Increased		
Breast cancer	miR-372, miR-101	Increased	Distinguishs between BC and benign tumors	[59]
	miR-24, miR-206, miR-1246, miR-373	Increased	BC detection	[60]
	miR-340-5p	Increased	Predicting recurrence	[61]
	miR-17-5p, miR-93-5p, miR-130a-3p	Decreased		
	miR-421, miR-128-2, miR-128-1	Increased	Predicts risks and adverse outcomes	[62]
	miR-30b	Decreased	Predicting recurrence	[63]
	miR-93	Increased	Ductal carcinoma in situ diagnosis	
	miR-16	Increased	Distinguishs ductal carcinoma in situ from health	
Ovarian cancer	hsa_circ_0061140	Increased	Promotes cell proliferation, invasion, apoptosis and EMT	[68]
	circRNA CDR1as	Decreased	Inhibits cell proliferation, migration and invasion	[69]
	circRNA CDR1as	Increased	Inhibits migration and invasion, promote cell proliferation and stagnation of G0/G1 cell cycle	[70]
	circ-SMAD7	Increased	Promotes cell proliferation, migration and invasion	[71]
	circHIPK3	Increased	Predicts risks and prognosis	[72]
	circLARP4	Decreased	Predicts risks and prognosis	[73]
	miR-34a	Increased	Predicts recurrence risks	[74]
	miR-21-5p	Increased	Diagnostic sign	[75]
	miR-29a-3P	Increased	Promotes the proliferation and metastasis of OC cells	
	miR-30a-5p	Increased	Diagnostic and therapeutic targets	
	Collagen type V alpha 2 chain (COL5A2)	Increased	Multiple diagnostic and prognostic markers of cancer	[64]
	lipoprotein lipase (LPL)		Unknown	
Cervical cancer	miR-221-3p	—	Promotes local angiogenesis	[78,79]
	miR-30d-5p, let-7d-3p	Decreased	Noninvasive screening diagnostic biomarkers for CC and its precursors	[80]
	miR-223	Increased	A key factor of the STAT3-miR-223-HMGCS1/TGFBR3 axis regulating the progression of CC	[81]
	LncRNA-EXOC7	—	Participates in cancer development and early diagnosis	[82]
	lncRNA HNF1A-AS1	—		[83]
Endometrial cancer	miR-148b	Decreased	Induction of endometrial cancer progression	[85]
	miR-320a	Decreased	Inhibits VEGFA expression and cell proliferation by targeting HIF1α	[86]
	Endothelial (CD144+), Monocytic (CD14+)	Increased	Related to histological grade and clinical stage of cancer	[91]
	miR-15a-5p, miR-106b-5p, miR107	Increased	correlates with tumor size and invasion depthEarly detection marker	[87]
	miR-423-3p, miR-195-5p, miR-20b-5p, miR-204-5p, miR-484, miR-143-3p	Increased	Early detection marker	[88]
	LGALS3BP	Increased	Endometrial carcinoma (EC) growth and angiogenesis	[84]
	ANXA2	Increased	Correlates with high grade, non-endometrioid subtype, advanced stage, and increased risks of recurrence	[92]
	miR-192-5p	Decreased	EC cell proliferationEMT	[89]
	miR-133a	Increased	EC and stromal cells communicate	[90]
Prostate cancer	miR-196a-5p, miR-501-3p	Decreased	Diagnostic marker	[93]
	miR-2909	Increased	Diagnostic markers, disease risk classification, Involved in metabolic and immune regulation by targeting MALT1, KLF4 and UPC2	[94]
	miR-1290, miR-375	Increased	Associated with cancer treatment and indicative of prognosis	[95]
	circ_0044516	Increased	PC cell survival and metastasis	[96]
	miR-1246	Increased	Associated with cell proliferation and EMT	[97]
	miR-940	Increased	Promotes osteogenic differentiation by targeting ARHGAP1 and FAM134A	[98]
	exosomal ephrinA2 and serum ephrinA2	Increased	Distinguishes PC patients from benign prostatic hyperplasia (BPH) patents	[99]

## 4. Application of Exosomes in the Treatment of Reproductive Diseases

Exosomes serve as stable and effective carriers and carry specific proteins, lipids, and genetic material, and thus, they provide a promising therapeutic transport vehicle for delivering bioactive substances to targeted tissues or organs [108]. Mesenchymal stem cells (MSCs) are the most widely used stem cells in the field of regenerative medicine. With strong proliferation ability in vitro, MSCs can be isolated from various tissues, such as bone marrow, fat, umbilical cord, and placenta [109]. As MSCs are believed to be the cells with the strongest exosome-producing capacity, MSC-derived exosomes (MSC-Exos) are widely used in the treatment of diseases related to the reproductive system.

### 4.1. The Function of MSC-Exos

MSC-Exos have the intrinsic potential to reduce inflammation and immune response similar to that of mesenchymal stem cells [110]. Studies have shown that adipose mesenchymal stem cell-derived exosomes play an immunomodulatory role by reducing the expression of interferon γ and transcription factors, up-regulating the expression of immunomodulatory cytokines, and reducing the local inflammatory response [111]. Moreover, MSC-Exos can enhance blood vessel growth and promote the proliferation of vascular endothelial cells, playing a crucial role in different physiological events, such as embryonic development, reproduction, and tissue regeneration and repair [112]. In addition, MSC-Exos can transfer and regulate proteins and miRNAs; up-regulate the expression of Bcl-2; down-regulate the expression of Bax; inhibit the apoptosis of bone cells, cardiomyocytes and epithelial cells; and promote tissue reconstruction [113,114,115].

### 4.2. The Application of MSC-Exos to Animal Models of Reproductive Diseases

In terms of ovarian insufficiency, MSC-Exos therapy alleviates ovarian injury in mice with ovarian insufficiency by regulating the SMAD signaling pathway [116]. In a chemotherapy-induced premature ovarian failure mouse model, human amniotic epithelium-derived exosomes inhibit ovarian granulosa cell apoptosis mainly by transferring miRNAs to restore ovarian function [14]. In polycystic ovary syndrome, miR-323-3p-modified adipose mesenchymal stem cell exosomes alleviate polycystic ovary syndrome by targeting programmed cell death protein 4 to inhibit granulosa cell apoptosis [24]. In rat models, bone marrow mesenchymal stem cell-derived exosomes improved ovarian structure and hormone levels [117]. In terms of intrauterine adhesions, umbilical cord mesenchymal stem cell exosome therapy can reduce the inflammatory response of the damaged endometrium, thereby promoting the regeneration of the damaged endometrium and reversing fibrosis [118]. Adipose mesenchymal stem cell-derived exosomes can not only improve the morphology of the injured endometrium in rats with intrauterine adhesion, but also improve the receptivity of the endometrium, thus improving the fertility of rats [119]. In terms of reproductive system tumors, adipose mesenchymal stem cell derived exosomes inhibit the proliferation and colony formation of A2780 and SKOV-3 cancer cells by blocking the cell cycle and activating mitochondria-mediated apoptosis signals, thus inhibiting ovarian cancer [120]. In addition, MSC-Exos are also used to guarantee embryonic development [121,122] and the vaginal injury repair process [115]. With the development of regenerative medicine, exosome therapy has brought new hope for patients with decreased ovarian function, polycystic ovary syndrome, and uterine adhesives. The therapeutic effect of MSC-Exos in animal models has been widely confirmed, but there is still a long way to go before clinical application.

## 5. Conclusions and Perspectives

As a new research hotspot, exosomes and their content have become a potential marker for reproductive disease diagnosis and treatment due to their extensive presence in vivo and their availability, which has a bright prospect in reproduction. However, the classification of exosomes is not clear, and there are no fully determined physical and biochemical criteria for the distinction between exosomes and microvesicles. In addition, the separation technology of exosomes is not perfect, which makes it difficult to obtain exosomes with high purity quickly, causing difficulties in subsequent detection and analysis. On the other hand, the detection system of exosomes has not been developed and mature, which is difficult to meet the detection of large clinical sample size. With the development of various highly sensitive and high-throughput analytical methods, the research and application of exosomes in the field of reproductive disease detection will be greatly accelerated.

## Figures and Tables

**Figure 1 vetsci-09-00706-f001:**
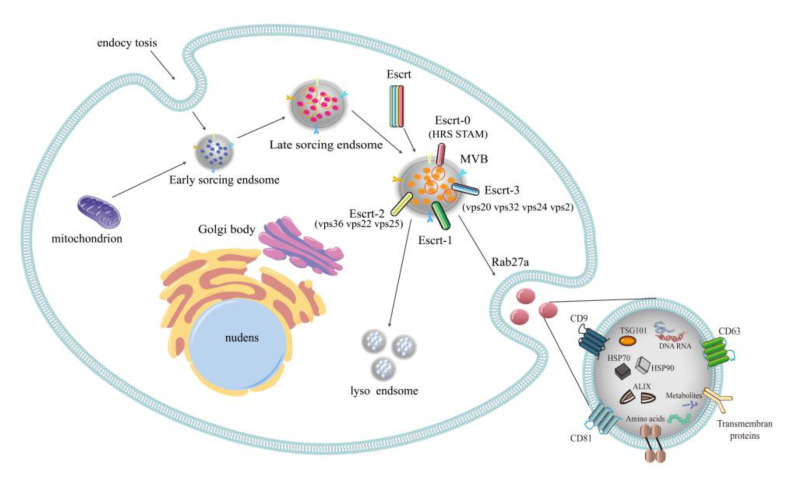
Production and secretion of exosomes. Exosome production involves double invagination of the plasma membrane and the formation of intracavitary vesicles (ILVs) and polyvesicles (MVBs). The first invagination of the plasma membrane forms early sorting endosomes (ESE), which then develops into mature late sorting endosomes (LSEs), and finally MVBs. MVBs contains multiple ILVs. MVBs are degraded by fusion with lysosomes or autophagosomes, or fuse with plasma membranes to release the ILVs contained in MVBs (exosomes). The secretion and transport of exosomes are mainly mediated by the Endosomal sorting complex Required for transport (ESCRT) on the MVBs membrane. ESCRT is mainly composed of ESCRT-0, -I, -II and -III complexes. Escrt-0 (HRS and STAM) regulates content aggregation through ubiquitination dependent pathways, ESCRT-I and ESCRT-II (Vps36, Vps22 and double-copy Vps25) induce bud formation. Escrt-III is composed of four core subunits: Vps20, Snf7 (Vps32), Vps24, and Vps2, as well as accessory proteins Did2, Vps60, and Ist1, which mainly promote membrane separation and vesicle cleavage. In addition, Rab27a and Rab27b, members of the Rab family GTPases, act on the docking of MVBs to the plasma membrane. ARF6 is a regulator of ILVs budding and exosome biosynthesis, which can promote the formation of ILVs.

## Data Availability

Not applicable.

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
