# Peer review of "The Role and Application of Exosomes and Their Cargos in Reproductive Diseases: A Systematic Review"

_vetsci, 2022, doi:10.3390/vetsci9120706_

Round 1

Reviewer 1 Report

The authors have submitted a manuscript reviewing "The role and application of exosomes and their cargos in repro-2 ductive diseases". However, at least a similar review artikel was published by Esfandyari et al  Int J Mol Sci (2021) 22(4):2165.

Since this is at least in part a systematic review, the authors should describe better their search/reviewing strategy. In Table 1 a very long and a bit confusing (in the lay-out) list of Exosomal markers for the diagnosis of reproductive diseases and their related functions is shown. However, most of the references in this list are not discussed in the main text (Refs. > #60, altogether more than 40 references). Referenced publications must be discussed, especially in a review article! In the text a Table 2 is mentioned (line 153) but this table is not available.

The origin of Fig. 1 remains unclear.

In general, the editor should decide if this type of manuscript is in general acceptable for the journal.

For the reviewer, this manuscript is in the current version not qualifying for publication.

Author Response

The authors have submitted a manuscript reviewing "The role and application of exosomes and their cargos in reproductive diseases". However, at least a similar review article was published by Esfandyari et al Int J Mol Sci (2021) 22(4):2165.

Response: Thank you very much for your valuable and constructive comments. We have read this paper carefully. The improvement of our manuscript is reflected in the following aspects: (1) We are supported by nearly 3 years of literature; (2) This paper is only female reproduction, we increase male reproductive disease, such asprostate cancer; (3) we add some different diagnostic and regulatory factors; (4) We increase the application research of mesenchymal stem cells derived exosomes.

Since this is at least in part a systematic review, the authors should describe better their search/reviewing strategy.

Response: Thank you very much for your valuable and constructive comments. We have added the section of search strategy in the manuscript. A systematic search was conducted in PubMed using the related keywords through September 2022. Related data from original researches and reviews were integrated to comprehensively review the current findings and understanding of exosomes and their cargos in reproductive diseases. In certain parts where human research was limited, data from animal studies were used for supplementation. The references are all scientific papers published in English.

In Table 1 a very long and a bit confusing (in the lay-out) list of Exosomal markers for the diagnosis of reproductive diseases and their related functions is shown. However, most of the references in this list are not discussed in the main text (Refs. > #60, altogether more than 40 references). Referenced publications must be discussed, especially in a review article!

Response: Thank you very much for your valuable and constructive comments. We have made necessary marks and explanations in the manuscript and table1. As we all know, Exosomes and their inclusions, such as miRNAs, are of various types and functions, so it is necessary to list their functions and regulatory pathways one by one. However, Exosomes and their inclusions are more commonly used as diagnostic markers in reproductive diseases. We review the representative exosome molecules and their roles in the manuscript. More exosome molecules and their functions are shown in Table 1. Readers can find exosome molecules corresponding to reproductive diseases they want to study more quickly and intuitively.

In the text a Table 2 is mentioned (line 153) but this table is not available.

Response: We are very sorry for the confusion caused by our writing mistake. There is no Table 2 in the manuscript, we have deleted it.

The origin of Fig. 1 remains unclear.

Response: Thank you very much for your comments. We've replaced it with a clearer picture.

Reviewer 2 Report

The article comprehensively explores a topical topic of definite interest in a clear and comprehensive manner.

- in figure one (line 67) is written early sorting and late sorting while in the drawing we find early sorcing and late sorcing- unifying 

Author Response

The article comprehensively explores a topical topic of definite interest in a clear and comprehensive manner. 

Response: Thank you very much for your comments.

- in figure one (line 67) is written early sorting and late sorting while in the drawing we find early sorcing and late sorcing- unifying.

Response: Thank you very much for your comments. We have revised it.

Reviewer 3 Report

The manuscript submitted for revision is the type of review manuscript which describe the exosomes cargo and its role in reproductive diseases. First of all, I'd like to clarify that exosomes are the type of vesicles; extracellular vesicles not substances (line 37-38).  It is not easy to prepare good review publication but definately it shoud be based on oryginal data (experimental data/publications) not other reviewd publication . Moreover, references 7 and 8 quoted in the paragraph : "1.2. Components of Exosomes" do not respond to the content presented therein.

Author Response

The manuscript submitted for revision is the type of review manuscript which describe the exosomes cargo and its role in reproductive diseases. First of all, I'd like to clarify that exosomes are the type of vesicles; extracellular vesicles not substances (line 37-38).

Response: Thank you very much for your comments. We have revised it.

It is not easy to prepare good review publication but definately it shoud be based on oryginal data (experimental data/publications) not other reviewd publication.

Response: Thank you very much for your comments. We tried our best to reorganize the structure, language and content of the manuscript.

Moreover, references 7 and 8 quoted in the paragraph : "1.2. Components of Exosomes" do not respond to the content presented therein.

Response: Thank you very much for your comments. We have revised it.

Round 2

Reviewer 1 Report

The revised version is better (and larger). The origin of Fig. 1 remains still unclear, the answer of the authors is rather the response to the question of another reviewer.

The authors answer now -as expected- that we are dealing with a systematic review. This must be mentioned in the Abstract and should be part of the Title. Due to the character of a systematic review the new paragraph 1 is important, however, a few references on the lay-out of systematic reviews and subsequently how these publications influenced the current manuscript should be added.

Table 1 should be clearer as mentioned in the first round of reviews

Author Response

The revised version is better (and larger). The origin of Fig. 1 remains still unclear, the answer of the authors is rather the response to the question of another reviewer.

Response: Thank you very much for your comments. We've replaced it with the origin picture.

The authors answer now -as expected- that we are dealing with a systematic review. This must be mentioned in the Abstract and should be part of the Title. Due to the character of a systematic review the new paragraph 1 is important, however, a few references on the lay-out of systematic reviews and subsequently how these publications influenced the current manuscript should be added.

Response: Thank you very much for your valuable and constructive comments. We have added the structure of the manuscript layout and the influence of the references cited on the manuscript. 

As follows:

Table of contents

  1. Search strategy
  2. II. Exosomes

(1) Formation of exosomes

(2) Multilaterality of exosomal contents

III. Exosomes in reproductive diseases

(1) Premature ovarian failuret

(2) Polycystic ovarian syndrome

(3) Preeclampsia

(4) Endometriosis

(5) Reproductive cancers

  1. Application of exosomes in the treatment of reproductive diseases

(1) The function of MSC-Exos

(2) The application of MSC-Exos to animal models of reproductive diseases

  1. Conclusions and Perspectives
  2. Author Contributions

VII. Funding

VIII. Competing interest

  1. References

This manuscript systematically collects relevant literature regarding the role of exosomes and their inclusions in the diagnosis, treatment and prognostic indication of reproductive diseases. The references cited in this manuscript are the in-depth analysis of exosomes and their inclusions that play a major role in reproductive diseases

Table 1 should be clearer as mentioned in the first round of reviews

Response: Thank you very much for your valuable and constructive comments. We have done our best to explain Table1 in detail in the manuscript.